# The Dual Impact of Digital Connectivity: Balancing Productivity and Well-Being in the Modern Workplace

**DOI:** 10.3390/ijerph22060845

**Published:** 2025-05-28

**Authors:** Giorgia Bondanini, Cristina Giovanelli, Nicola Mucci, Gabriele Giorgi

**Affiliations:** 1Business@Health Laboratory, European University of Rome, 00190 Rome, Italy; cristina.giovanelli@unier.it (C.G.); gabriele.giorgi@unier.it (G.G.); 2Department of Human Sciences, European University of Rome, 00190 Rome, Italy; 3School of Occupational Medicine, University of Florence, Largo Brambilla, 3, 50134 Florence, Italy; nicola.mucci@unifi.it; 4Department of Experimental and Clinical Medicine, University of Florence, Largo Brambilla, 3, 50134 Florence, Italy

**Keywords:** digital connectivity, smart working, well-being, remote work, digital overload, occupational health

## Abstract

Background: Digital connectivity is essential in modern work environments, enhancing productivity and communication. However, its rapid expansion post-COVID-19 raises concerns about burnout, digital fatigue, and work-related stress. Objective: This PRISMA-based systematic review examines the benefits and challenges of digital work, assessing its impact on occupational health and proposing mitigation strategies. Methods: A systematic search of PubMed, Google Scholar, Scopus, and Cochrane Library identified 40 peer-reviewed studies published since 2020, focusing on digital connectivity, remote work, and employee well-being. Studies on purely technological aspects were excluded. Results: While digital tools improve efficiency and flexibility, they also increase workload, cognitive overload, and stress. Prolonged screen exposure contributes to mental exhaustion and sleep disturbances. Limited digital infrastructure further exacerbates productivity barriers. Conclusions: Digital connectivity offers both opportunities and risks. Organizations should implement structured policies such as offline hours, digital detox initiatives, and mental health support to sustain productivity and well-being. Future research should explore sector-specific interventions and long-term impacts of digital work practices.

## 1. Introduction

So far, the deepened blending of digital connectivity into the workplaces today has changed how individuals engage socially, work in collaboration, and perform work that shapes productivity. The move toward remote work—certainly hastened by the COVID-19 pandemic—has disrupted age-old constructs of work, enabling people to do their jobs from just about anywhere. This digital transformation brought several advantages: greater flexibility, improved communication, and enhanced operational efficiency [1]. Moreover, organizations have taken advantage of digital solutions to optimize workflows, automate processes, and enable worldwide collaboration. However, this transition has come with significant challenges in its own right, affecting employee well-being and mental health [2]. A commonly discussed topic of concern surrounding digital connectivity is work–life balance. Smart working definitely enables employees to do without crammed travel to and from the office, allowing them to gain more control over their schedule, but it also fuzzes the boundary between personal and professional [3]. The ability to easily access work-related communications outside of traditional working hours has resulted in an “always on” culture and increased stress levels, as well as more difficulty detaching from work-related commitments. Galanti et al.’s (2021) study suggests that overuse of digital tools, especially videoconferencing and instant messaging platforms, presents a risk of cognitive overload, digital fatigue, and social isolation [4]. It has also been apparent that the mental health ramifications of digital connectivity can manifest themselves through burnout and sleep disturbance [5]. The demand for constant accessibility, even with extended screen time, has raised stress levels among employees across a range of industries. According to a study conducted by Abdeen and Khalil in 2023, it was found that increased use of digital work platforms where more digital work is shared had significant implications that led to lower productivity, job dissatisfaction, and higher turnover intentions [6]. Some employees reported experiencing increased levels of anxiety and depression resulting from the barrage of digital communication and virtual collaboration [7]. Nevertheless, despite these factors being relevant information, remote working and with that, digital connectedness, are central in any contemporary company’s prominence. So, many organizations are starting to look for solutions to mitigate the negative effects of excessive digital screen time. Structured remote work policies, scheduled breaks, and ergonomic recommendations are just some of the strategies being put in place to ensure productivity does not come at the expense of health and well-being. Moreover, the development of artificial intelligence and automation may help to relieve workload pressures by streamlining repetitive tasks or optimizing time management [8].

A socio-economic issue that needs to be tackled with digital connectivity is the gap between those with access and those without access to information systems; this situation is called the digital divide. Workers with steady online access and high-end equipment enjoy consistent virtual teamwork; however, wealthy employees flourish while employees in poor regions trail behind. This inequality can imply further effects on the satisfaction of employees, such as meeting career and growth perspectives, and also can have an overall impact on their performance in the workplace, thus raising the demand for organizations to adjust this access in order to ensure equal places due to the existing digital infrastructure and resources [9].

However, from the broader organizational filter, the adoption of technology has driven profound alterations in ways of working, inducing firms to reconsider their management and employee participation strategies. It is up to employers to find that balance of how to encourage innovation on behalf of their workforce while managing the various potential negatives of working in remote and hybrid work environments. In an effort to mitigate the challenges brought about by excessive digital connectivity, organizations have attempted to intervene with wellness initiatives, digital detox efforts, and mental health resources [10].

From a psychological standpoint, the digital connectivity challenge facing employees is another area of concern. The trend of remote working and digital collaboration is leaving many professionals feeling lonely and disengaged. Indeed, we have swapped social face time for streamlined interfaces, with exchanges now taking place in a world that until very recently was face to face, making us efficient and hyper-connected but perhaps less emotionally parabolic. Yusuf et al. (2023) found that regular virtual events can help team cohesion but also cause video call fatigue and, in some instances, diminished engagement overall [11].

On the other hand, the medical aspect also has a lot of concerns, as prolonged exposure to digital screens and electronic devices has been linked to physical health issues, such as eye strain, musculoskeletal problems, and sleep disturbances. Moreover, the blue light from screens can impact circadian rhythms, making it hard to fall asleep and reducing sleep quality [12].

This systematic review addresses the effect of digital connectivity on occupational well-being, specifically on smart working, burnout, and related health issues. This study seeks to provide a comprehensive and integrative understanding of the advantages and disadvantages of digital environments and possible approaches to a sustainable and employee-centric digital work atmosphere through a literature review [2].

## 2. Materials and Methods

This systematic review followed the PRISMA checklist 2020 guidelines and was registered on PROSPERO ID 1040461 [13].

### 2.1. Inclusion and Exclusion Criteria

In order to achieve a systematic and reproducible search for studies, we defined the inclusion and exclusion criteria presented below. Studies that were included in the review were those that (1) came from peer-reviewed studies; (2) were published in scientific journals; (3) presented clear methodologies; these could include observational, experimental, and systematic reviews and meta-analysis study designs; (4) evaluated effects and impacts from the increased digital connectivity related to burnout, insomnia, smart working, work–life balance, and work-related stress; and (5) were published from 2020, in order to capture articles throughout the pandemic. In the selection of studies, the exclusion criteria eliminated duplicates as a result of searching on different databases; studies that focused only on explaining the technical aspects of digital connectivity, without elucidating its effects on health or well-being; studies not available in English or Italian; studies not consultable in full text; and articles of opinion where the methodology of the study was not clear. Applying these selection criteria, 40 studies were ultimately included in the final analysis.

### 2.2. Data Extraction

Four major academic databases, namely PubMed, Google Scholar, Scopus, and Cochrane Library, were utilized to identify relevant studies. The search strategy had the following keywords: digital connectivity, smart working, insomnia, burnout, well-being, social impact, COVID-19 pandemic, digital work, digital transformation, and work–life balance. As summarized in Table 1, 500 studies were initially retrieved from these databases. After the removal of duplicates and irrelevant articles, 200 studies were left for additional screening. A total of 100 studies were subjected to full-text evaluation, with 40 studies finally included in this meta-analysis. Furthermore, additional relevant information can be obtained by consulting Table A1, which contains the search string.

### 2.3. Results Interpretation

The identified studies were systematically examined and classified according to their design and thematic emphasis. We organized the study data by date until October 2023, and by topic, such as the source of occupational burnout and stress related to remote working, the relationship between digital connectivity and disrupted sleep such as insomnia, the effect of remote working on work–life balance and generalized well-being, and videoconferencing and digital communication tools’ psychological effects on employees. The studies synthesized findings from experimental and longitudinal studies to understand trends and causality, while systematic reviews and meta-analytic findings took a broader view of the topic.

### 2.4. Quality Assessment

A quality assessment was carried out based on criteria defined to ensure the reliability and validity of inducted studies. The evaluation focused on study design and methodological rigor, clarity and transparency in data collection and analysis, relevance to the research objectives, and robustness of statistical analyses and reported outcomes. Studies with poor methodologies, vague data, or no definitive results were eliminated to ensure that only high-quality studies were included in the final synthesis.

No formal risk of bias tool was applied due to the heterogeneity of study designs. Instead, studies were assessed narratively for clarity of aims, sample relevance, and data transparency.

## 3. Results

Finally, 40 studies met the inclusion criteria and were analyzed. A flow diagram following the principles of PRISMA, showing how we selected papers for further evaluation, is presented in Figure 1.

The results of these studies highlighted the benefits of digital connectivity, such as greater work flexibility and productivity, alongside negative effects, including greater digital fatigue, burnout, and sleep-related impairments. According to studies, remote working was adopted by employees with appropriate digital tools, enhancing efficiency and work–life balance. Yet overexposure to digital screens and long, responsive virtual meetings were often correlated with cognitive overload, eye strain, and increased stress levels. Additionally, employee satisfaction was swayed by digital access and infrastructure, as employees with stable Wi-Fi connectivity were 59 per cent more productive than those with interruptions.

So, the results varied widely between industries—a reminder to consider job roles and sector-specific demands. For instance, studies targeting healthcare professionals reported high levels of stress and burnout connected to ongoing digital communication, while studies that examined corporate employees revealed that flexible digital work arrangements resulted in better well-being. Also, workplace policies, like breaks from digital devices and ergonomic guidelines, proved to be effective in minimizing the adverse effects of extensive digital connectedness.

Table 2 depicts a summary of the selected studies, their main findings, and thematic focuses. As a systematic review, it highlights both the benefits and challenges around digital connectivity we now frequently experience in contemporary ways of working, and emphasizes the promise or need for balanced digital working strategies to promote employee well-being and productivity.

## 4. Discussion

This review demonstrates the complex, dual, and multifaceted nature of digital connectivity in contemporary space. It has been reported that numerous digital tools have proven beneficial, as they provide increased efficiency, cost savings, improved communication, etc., but this raises concerns related to employee well-being due to increased digitalization. While working from home, and hybrid working is more flexible than in-office working, digital connectivity has some side effects such as work–life imbalance, higher workload expectations, and loneliness [15]. The growth of digital connectivity is unprecedented, rising rapidly with the dawn of the post-COVID-19 era. As the pandemic has pushed organizations towards digitalization, scientists are just starting to glimpse how the architectural implications of this phenomenon will unfold over time. The former suggests the growing significance of digital fatigue, social isolation, and blurring of work–life balance as issues of concern. With the reliance on digital tools only set to escalate, there is a cost in terms of both talent and culture, and monitoring (and evaluating) these trends should be a key priority before they become entrenched in the corporate culture [46].

Burnout, one of the key consequences of extended digital connectivity, has become a leading challenge every organization must tackle. Excessive employee engagement in video calls, instant messaging, and other online activities leads to them becoming mentally exhausted and unproductive at work. From these insights, companies should implement practices that mitigate the adverse effects of digital burnout, such as mandatory breaks, establishing meeting limits, and promoting interpersonal communication styles. Additionally, organizations could explore innovative approaches to enhance employee engagement and morale, such as virtual team-building activities that foster connection and collaboration among remote workers [2,4].

Furthermore, company policies need to evolve to recognize the fallout of digital burnout. A trend that some organizations are already embracing is structured work schedules, where employees observe “offline hours” each day (during which they are not expected to respond to emails or attend meetings in virtual rooms). Creating a culture of digital well-being, where employees feel less likely to fear losing their job for not being available, can significantly improve job satisfaction and overall mental well-being. This cultural shift can also encourage employees to prioritize their mental health, ultimately leading to a more productive and satisfied workforce [30].

Other factors not to be overlooked are digital capital and digital diversity. As we have seen above, employees with poor internet infrastructure or limited access to digital tools struggle a lot to keep up with productivity and all the technological work that is conducted around the world today. To mitigate the risk of disparities in access to digital resources, organizations and policy makers should prioritize equitable access to the digital economy. This includes investing in infrastructure improvements and providing training programs to help all employees develop the necessary digital skills to thrive in a tech-driven environment. By fostering inclusivity in digital access, organizations can harness the full potential of their workforce and drive innovation [40,41].

## 5. Conclusions

This systematic review of the scientific literature emphasizes the dual-sided nature of connectivity in today’s digital work environments. As a digital work solution, it presents the transformation digital tools have made for the workplace, providing collaboration and efficiency opportunities; but at the end of the day, this review is an argument for fast adopters of digital work practices to implement a methodical approach with a certain set of guidelines to mitigate the risks of burnout, digital fatigue, and work-related stress.

New technologies have changed roles and work structures in such a way that requiring adaptability and digital literacy has become a must in the new workforce. In this new professional and global environment characterized by ever deeper embedding links between work and the citizen, employees need to balance the gregariousness of time tracking and virtual communication; it is thus paramount to learn how to successfully establish the border between work and personal life. Indeed, the reach of digital connectedness goes well beyond each worker alone at their desk; it extends to the nature of organizations themselves—affecting team cohesion, leadership styles, and potentially even the global culture of business. Businesses that are unable to adjust to these shifts would face decreased employee engagement and increased turnover rates, which would, in turn, lead to decreased productivity.

From a psychological standpoint, the effects of digital connectivity have demonstrated that excessive exposure to screens can lead to cognitive overload, stress, and emotional exhaustion. Real-time communication tools have birthed an expectation for businesses that we are always on, breaking the natural flow of work patterns, resulting in broken focus and less job satisfaction. Consequently, it is pertinent for organizations to identify these risks and adopt preventative measures to mitigate them with well-defined workplace policies and interventions by developing structured digital working practices, promoting digital detox initiatives, and facilitating peer interactions. Moreover, developing a corporate culture that understands the significance of mental health support and work–life balance will be key to achieving sustainable prosperity in a digitally connected world. Indeed, investing in employee training programs on digital well-being, time management, and stress reduction can help foster these attributes in the workforce even more.

However, this review presents several limitations that should be acknowledged. One key limitation is that this study presents a high heterogeneity, particularly in terms of study design, population characteristics, and outcome measures, which limits both the comparability and generalizability of the findings. Additional potential biases may be observed, such as the limited inclusion of gray literature and non-English language sources, which may have excluded relevant but unpublished or regionally disseminated studies. An additional limitation concerns the use of formal tools, as no formal risk of bias tool was applied due to the heterogeneity of the study designs. Instead, studies were assessed narratively for clarity of aims, sample relevance, and data transparency.

Data: Further research can also explore ways to measure the effectiveness of innovations produced within the virtual economy. Such analyses will be essential to shaping effective corporate policies. Companies need to constantly rethink the strategies around digital work as they respond to the evolving needs of their employees.

Moreover, as AI-driven tools are becoming more and more embedded in digital workspaces, it is critical for organizations to reflect on how automation and algorithmic management may influence employee autonomy, skills, and psychological safety. These dimensions should not be overlooked when designing sustainable and human-centered digital transformation strategies.

Those organizations that embrace a holistic approach to well-being as part of their digital transformation programs will fare better in this brave new world. Realigning the unintended effects of digital work will not only help employees be even happier at their jobs but will also positively impact the future of organizations at a macro level.

## Figures and Tables

**Figure 1 ijerph-22-00845-f001:**
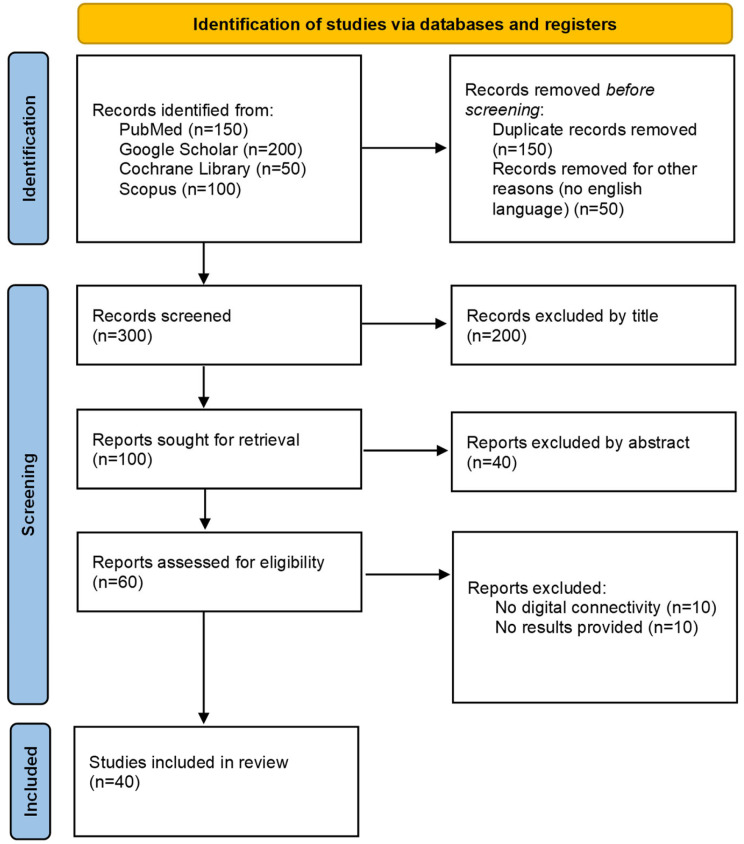
PRISMA flow diagram.

**Table 1 ijerph-22-00845-t001:** Study selection.

Phase	Number of Studies	Reason for Selection or Exclusion
Studies initially identified	~500	Retrieved from Google Scholar, PubMed, Scopus, and Cochrane Library.
Studies after duplicate removal and screening	~200	Exclusion of duplicates and non-relevant articles.
Full-text studies assessed for eligibility	~100	Retained based on methodological rigor and relevance.
Final studies included	~40	High-quality studies used for PRISMA-based review.

**Table 2 ijerph-22-00845-t002:** Summary of the selected studies.

Author(s)	Year	Study Objective	Methodology	Main Findings
Nadler, R. [14]	2020	Examine the impact of videoconferencing on communication and cognitive fatigue.	Literature review.	Videoconferencing can increase cognitive fatigue due to the lack of non-verbal cues and the extra effort needed to process information.
Sasaki, N. et al. [1]	2020	Investigate the impact of remote work on employees’ mental health during the COVID-19 pandemic in Japan.	Cross-sectional study with online questionnaires.	Remote work is associated with lower psychological stress levels compared to in-office work during the pandemic.
Majumdar, P. et al. [15]	2020	Examine screen exposure and mental fatigue in remote workers.	Experimental study measuring cognitive fatigue.	Extended screen time significantly increases mental fatigue.
Bailenson, J.N. [16]	2021	Analyze the causes of "Zoom fatigue" and propose solutions to mitigate it.	Theoretical analysis based on previous studies.	Excessive close-up eye contact, constant self-view, and restricted mobility contribute to videoconference fatigue.
Fauville, G. et al. [17]	2021	Develop and validate the "Zoom Exhaustion & Fatigue Scale" (ZEF) to measure videoconference-related fatigue.	Empirical study with participants using videoconferencing tools.	The ZEF scale is a valid tool for measuring fatigue associated with video conferencing use.
Bennett, A.A. et al. [18]	2021	Explore changes in fatigue after videoconference meetings during the COVID-19 pandemic.	Longitudinal study with pre- and post-meeting surveys.	Videoconference meetings are linked to increased fatigue, especially when prolonged without breaks.
Okabe-Miyamoto, K. et al. [19]	2021	Examine the effect of constant connectivity on employee productivity during the pandemic.	Quantitative study with online surveys.	Costant connecivity is associated with a decrease in perceived productivity.
Zhang, Y. et al. [20]	2021	Investigate the impact of videoconferencing on workers’ mental health.	Cross-sectional study with psychological questionnaires.	Frequent videoconferencing is correlated with symptoms of anxiety and depression.
Nguyen, T. et al. [10]	2021	Assess the effect of videoconferencing on stress management in remote teams.	Qualitative study with manager interviews.	Videoconferencing can be stressful for remote teams, requiring specific management strategies.
Ivanov, D. et al. [21]	2021	Investigate the impact of videoconfere on information security in organizations.	Case study with security incident analysis.	Videoconferencing can increase information security risks if not managed properly.
Shockley, K.M. et al. [2]	2021	Investigate the effect of remote work on employee well-being during the COVID-19 pandemic.	Cross-sectional study with online surveys.	Remote work has mixed effects on well-being, improving work-life balance for some but increasing burnout for others.
Galanti, T. et al. [4]	2021	Analyze the effect of remote work on employee well-being and work-life balance during the COVID-19 pandemic.	Quantitative study with employee surveys.	Remote work improved work-life balance but also increased feelings of isolation and burnout.
Roberts, J.K. et al. [22]	2021	Investigate the effect of constant connectivity on cognitive load.	Experimental study with cognitive task performance.	Constant connectivity increases cognitive load compared to face-to-face communication.
Dresp-Langley, B. et al. [5]	2022	Analyze the impact of prolonged digital device use on sleep during the pandemic.	Observational study with sleep monitoring.	Increased digital device use, especially before bedtime, is associated with reduced sleep quality and higher insomnia rates.
Bleakley, A. et al. [23]	2022	Investigate how videoconferencing affects team collaboration.	Experimental study with team tasks.	Videoconferencing reduces collaboration efficiency due to delayed communication.
Battisti, E. et al. [24]	2021	Examine the effect of digital connectivity on employees’ time management.	Quantitative study with work-time analysis.	Digital connectivity can lead to inefficient time management and work overload.
Johnson, K.J. et al. [25]	2021	Examine digital platform use and sleep disorders during COVID-19.	Prospective cohort study.	Prolonged digital platform use was associated with insomnia.
Chanana, N. et al. [26]	2021	Examining the role of hyper-connectivity to digital devices in employee engagement.	Survey-based study with remote workers.	Hyper-connectivity can reduce engagement due to cognitive overload.
Döring, N. et al. [27]	2022	Examine the association between hyper-connectivity use and eye fatigue.	Experimental study with ocular measurements.	Prolonged hyper-connectivity is linked to symptoms of eye strain, highlighting the need for regular breaks.
Alvarez-Gutierrez, L. et al. [28]	2022	Examine the effect of digital connectivity on leadership effectiveness.	Interviews with remote team leaders.	Leaders find it harder to build rapport and trust through videoconferencing.
Raake, A. et al. [29]	2022	Explore coping strategies for videoconferencing fatigue.	Meta-analysis of existing studies.	Strategies like reducing screen time and incorporating breaks help mitigate fatigue.
Van Zoonen, W. et al. [30]	2022	Analyze the psychological effects of prolonged virtual communication.	Longitudinal study with mental health assessments.	Prolonged virtual communication is linked to higher stress and anxiety levels.
Park, E. S. et al. [31]	2023	Investigate the impact of remote work policies on employee satisfaction.	Large-scale employee survey.	Flexible remote work policies increase satisfaction, but excessive videoconferencing negates benefits.
Tsai, H. H.. et al. [32]	2023	Analyze the relationship between digital connectivity and social isolation.	Longitudinal study.	Digital connectivity can increase feelings of social isolation.
Costin, A. et al. [33]	2023	Assess the impact of digital connectivity on productivity and employee well-being.	Longitudinal study.	Effective digital connectivity was crucial for productivity but led to burnout and digital fatigue.
Thompson, S.M. et al. [34]	2022	Analyze the impact of hyper-connectivity on job satisfaction.	Longitudinal study.	Excessive hyper-connectivity can reduce job satisfaction over time.
Evans, H. et al. [7]	2023	Assess the impact of virtual backgrounds on videoconferencing fatigue.	Experimental study.	Virtual backgrounds can reduce visual fatigue but may increase cognitive strain.
Bailey, D. R. et al. [35]	2023	Analyze videoconferencing’s effect on learning engagement.	Longitudinal study.	Videoconferencing can decrease student engagement compared to in-person learning.
Martinez et al. [36]	2023	Analyze the impact of remote work on mental health among teachers.	Cross-sectional study.	Remote teachers reported higher burnout than in-person teachers.
Choung, Y. et al. [37]	2023	Investigate digital connectivity’s effect on well-being in the financial sector.	Mixed-method study.	Stable digital connectivity improved productivity but increased stress.
Garcia-Perez, A. et al. [8]	2023	Explore digital connectivity’s impact on healthcare workers’ well-being.	Qualitative study.	Digital connectivity made separating work from life harder, increasing stress.
Palos-Sanchez, P. R. et al. [38]	2023	Investigate videoconferencing’s effect on team cohesion.	Mixed-method study.	Videoconferencing can reduce team trust and cohesion.
Chen, W. H. et al. [39]	2023	Assess the impact of virtual team meetings on innovation.	Case study with R&D teams.	Virtual meetings limit spontaneous idea exchange, reducing innovation.
Deng, H. et al. [40]	2023	Assess digital communication’s impact on job performance.	Mixed-method study with employee feedback.	Digital communication tools improve efficiency but increase stress.
Charoenporn, V. et al. [41]	2023	Evaluate how AI-driven tools can reduce videoconferencing fatigue.	Experimental study using AI-assisted communication.	AI tools that summarize meetings reduce cognitive strain and improve efficiency.
Luebstorf, S. et al. [42]	2023	Examine strategies to reduce digital fatigue in remote workers.	Experimental study.	Regular breaks and adjusting screen time significantly reduce digital fatigue.
Štukovnik, V. et al. [43]	2024	Assess the effect of hyper-connectivity on employees’ sleep quality.	Observational study.	Hyper-connectivity is associated with reduced sleep quality.
Xiao, T. et al. [44]	2024	Examine digital device use and sleep disorders.	Observational study.	Prolonged device use is correlated with sleep disturbances.
Delgado, K. et al. [45]	2024	Analyze videoconferencing’s impact on professional training.	Qualitative study.	Videoconferencing is less effective than in-person training.
Brown, T. et al. [12]	2024	Analyze the impact of hyper-connectivity on posture and musculoskeletal pain.	Ergonomic study with postural assessments.	Hyper-connectivity is linked to poor posture and increased musculoskeletal pain.

A meta-analysis was not conducted due to the heterogeneity among the included studies in terms of design, outcome measures, and populations investigated. Therefore, a narrative synthesis was deemed more appropriate to preserve the interpretative validity of the findings.

## Data Availability

Data is contained within the article.

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
