# Peer review of "The Dual Impact of Digital Connectivity: Balancing Productivity and Well-Being in the Modern Workplace"

_ijerph, 2025, doi:10.3390/ijerph22060845_

Round 1

Reviewer 1 Report

Comments and Suggestions for Authors

The paper analyzes a very interesting and current matter, which is the impact of digital connectivity on workers' well-being and productivity. The analysis of the existing literature is extensive and thorough, and the methodology for the selection of the analyzed papers is well explained and detailed.

Nevertheless, in my opinion, the discussion on the 40 selected papers should be richer. The overall conclusion is that digital connectivity offers both opportunities and risks and that organizations should implement structured policies to offset these risks. Nevertheless, despite the accuracy of the conclusion, it is a well-established conclusion in the literature (described by some authors as a paradox).

The discussion, I believe, could be richer and more detailed. The summary table included in the paper is very interesting, but my opinion is that the discussion should include more detail of the different papers studied: the positive and negative effects of digital connectivity, the constraints or limitations of these positive and negative effects (for example, some authors have identified that negative effects of digital connectivity can be "compensated" or are reduced when workers benefit from flexible working time) and possible policy implications identified in the different papers.

Reviewer 2 Report

Comments and Suggestions for Authors
  1. Summary and overall evaluation:

The manuscript presents a systematic review on the effects of digital connectivity on productivity and well-being in the workplace. The topic is timely, especially in the context of accelerated digitalisation and remote work trends in the post-pandemic era, and it fits within the journal’s scope.

The review includes a PRISMA flow diagram (Figure 1) and a summary table of included studies (Table 2). The structure is aligned with a systematic review format. However, the manuscript requires revisions to improve methodological reporting, organisation of results, and editorial quality. Table 2 should be revised to conform to established standards for data presentation in systematic reviews. Additional clarification of procedures and findings is also needed.

  1. Major comments

PRISMA

  • The manuscript includes a PRISMA flow diagram. Authors should provide a PRISMA checklist as supplementary material.
  • Protocol registration is not mentioned. If not registered, the authors should state this.
  • The review does not describe whether a quality appraisal of included studies was conducted. A recognised tool (e.g., MMAT, JBI) should be used and reported.

Abstract

  • The abstract does not follow the suggested logical order (Background → Methods → Results → Conclusions). It should include the number of studies reviewed and a summary of the main findings.

Introduction

  • The Introduction references the study as a systematic review and includes a statement of purpose (lines 97–101). However, the aim could be made clearer by rephrasing and presenting it as a standalone sentence at the end of the Introduction.
  • The phrasing (e.g., “to obtain a holistic overview”) should be revised using more concise and formal review terminology.
  • The scope of the review (e.g., target populations, work sectors, outcomes) should also be briefly clarified.

Results

  • Table 2 provides a list of included studies but lacks consistency in structure. Key data such as study design, population, sample size, setting, and outcomes are not presented uniformly.
  • The format of Table 2 limits comparison between studies. A more structured layout with defined columns would help.
  • The text does not explain why meta-analysis was not performed. If not feasible, the authors should state the reason.

Discussion

  • The discussion does not sufficiently analyse differences or similarities among the included studies and lacks depth in terms of critical comparison.
  • The implications for policy, practice, or further research are not clearly developed.
  • Limitations of the review are mentioned but should address study heterogeneity, lack of longitudinal data, and potential biases.

Conclusions

  • The conclusion restates earlier sections but does not fully summarise the contributions of the review.
  • Practical implications and recommendations for future research or workplace policy should be briefly added.
  1. Minor comments

Title and Keywords

  • The title is relevant and concise.
  • The manuscript includes three keywords, which meets the minimum requirement. However, MDPI recommends including three to ten keywords. Expanding the list will improve indexing. Suggested additions: remote work, digital overload, occupational health.

Figure 1 (PRISMA Flow Diagram)

  • Figure 1 is appropriate and well-structured.
  • Ensure the image is submitted in a high-resolution format.

Table 2 (Summary of included studies)

  • Table 2 lacks a descriptive caption (line 173).
  • Table 2 should be revised to conform to established standards for data presentation in systematic reviews.
  • Suggested columns: Author(s) and Year, Study Design, Population/Setting, Digital Exposure, Outcome Type, Key Findings, Reference.
  • Studies may also be grouped by thematic focus (e.g., productivity vs. well-being).

Ethics

  • The manuscript includes statements on ethical review and informed consent, which are not applicable to a systematic review. These should be removed or rephrased appropriately.

References

  • Ensure all references comply with MDPI formatting requirements.

Round 2

Reviewer 2 Report

Comments and Suggestions for Authors
  1. General comments

The authors have responded constructively to the previous peer review and implemented substantial improvements across the manuscript. The revised manuscript is now more methodologically transparent and conceptually coherent. However, revisions are still necessary, particularly regarding the PRISMA checklist submission, clarification of the quality appraisal method, and search strategy reporting.

  1. Major comments

PRISMA checklist
Although a PRISMA flow diagram is included and search strings are provided in Supplementary Table 1, the PRISMA 2020 checklist itself is not included. While it is not strictly mandatory, submitting a PRISMA 2020 checklist as a separate supplementary document is strongly recommended and is often expected when you submit a systematic review to a journal.

Quality appraisal
Clarification about the statement: “The review does not describe whether a quality appraisal of included studies was conducted. A recognised tool (e.g., MMAT, JBI) should be used and reported.”

The manuscript includes a narrative description of quality criteria in section 2.4. However, it is unclear whether a formal quality assessment tool was applied. In systematic reviews, critical appraisal typically involves the use of established tools such as:
- JBI Checklists (for study-specific appraisal)
- MMAT (for mixed-methods reviews)
- CASP (for qualitative studies)

Based on the manuscript, which includes qualitative, quantitative, and mixed-methods studies with diverse designs (e.g., cross-sectional surveys, interviews, literature reviews, experimental studies), the most appropriate quality appraisal tool is the Mixed Methods Appraisal Tool (MMAT).
If no formal tool was used, the authors should explicitly state this in the Methods and justify the decision in the Limitations section. For example: “No formal risk of bias tool was applied due to the heterogeneity of study designs. Instead, studies were assessed narratively for clarity of aims, sample relevance, and data transparency.”

Search strategy reporting
The authors included the search strings in a supplementary file. However, key metadata is still missing and should be added in the Methods section or as part of the supplementary material:

  • Search dates.
  • Filters applied (e.g., date limits, publication type).
  • Database-specific notes (e.g., only articles, reviews, full text).

  1. Minor comments
  • Table 1 lacks a descriptive caption (line 129).
  • Table 2 still contains minor issues (e.g., truncated? words, like “videoconfere” - Ivanov, D. et al. [20]). A final proofreading pass is recommended to ensure consistency.
  • The manuscript would benefit from careful proofreading to correct minor grammatical and stylistic issues throughout.
